# The Application of Machine Learning Models Based on Leaf Spectral Reflectance for Estimating the Nitrogen Nutrient Index in Maize

Bo Chen [1,2,3], Xianju Lu [2,3], Shuan Yu [2,3], Shenghao Gu [2,3], Guanmin Huang [2,3], Xinyu Guo [2,3,*] and Chunjiang Zhao [1,2,3,*]

1 College of Resources and Environment, Jilin Agricultural University, Changchun 130118, China
2 Beijing Key Laboratory of Digital Plant, National Engineering Research Center for Information Technology in Agriculture, Beijing 100097, China
3 Information Technology Research Center, Beijing Academy of Agriculture and Forestry Sciences, Beijing 100097, China
* Correspondence: guoxy73@163.com (X.G.); zhaocj@nercita.org.cn (C.Z.)

**Abstract:** Non-destructive acquisition and accurate real-time assessment of nitrogen (N) nutritional status are crucial for nitrogen management and yield prediction in maize production. The objective of this study was to develop a method for estimating the nitrogen nutrient index (NNI) of maize using in situ leaf spectroscopy. Field trials with six nitrogen fertilizer levels (0, 75, 150, 225, 300, and 375 kg N ha$^{-1}$) were performed using eight summer maize cultivars. The leaf reflectance spectrum was acquired at different growth stages, with simultaneous measurements of leaf nitrogen content (LNC) and leaf dry matter (LDW). The competitive adaptive reweighted sampling (CARS) algorithm was used to screen the raw spectrum's effective bands related to the NNI during the maize critical growth period (from the 12th fully expanded leaf stage to the milk ripening stage). Three machine learning methods—partial least squares (PLS), artificial neural networks (ANN), and support vector machines (SVM)—were used to validate the NNI estimation model. These methods indicated that the NNI first increased and then decreased (from the 12th fully expanded leaf stage to the milk ripening stage) and was positively correlated with nitrogen application. The results showed that combining effective bands and PLS (CARS-PLS) achieved the best model for NNI estimation, which yielded the highest coefficient of determination ($R^2_{val}$), 0.925, and the lowest root mean square error (RMSE$_{val}$), 0.068, followed by the CARS-SVM model ($R^2_{val}$, 0.895; RMSE$_{val}$, 0.081), and the CARS-ANN model ($R^2_{val}$, 0.814; RMSE$_{val}$, 0.108), which performed the worst. The CARS-PLS model was used to successfully predict the variation in the NNI among cultivars and different growth stages. The estimated $R^2$ of eight cultivars by the NNI was between 0.86 and 0.97; the estimated $R^2$ of the NNI at different growth stages was between 0.92 and 0.94. The overall results indicated that the CARS-PLS allows for rapid, accurate, and non-destructive estimation of the NNI during maize growth, providing an efficient tool for accurately monitoring nitrogen nutrition.

**Keywords:** summer maize; nitrogen nutrient index; leaf spectral reflection; effective band; machine learning

## 1. Introduction

Nitrogen, as one of the most important nutrients in crops, greatly influences crops' growth and development, quality, and yield formation and is the main element that affects crop development in agricultural production [1,2]. Insufficient nitrogen results in smaller crop leaves, lower chlorophyll and protein content, reduced dry matter accumulation, and reduced grain yield and quality [3]. To ensure crop yields, growers often apply excess nitrogen fertilizer in the field [4]. However, excessive fertilizer application can cause environmental and ecological problems, such as increased greenhouse gas emissions,

groundwater pollution, and surface water eutrophication [5]. Therefore, an accurate and timely understanding of the nitrogen nutritional status of crops and optimization of nitrogen fertilizer management measures can improve crops' nitrogen use efficiency and the ecological benefits of crop production [6].

The nitrogen nutrient index (NNI), which is the ratio of the plant's actual nitrogen concentration (PNC) to the plant's critical nitrogen concentration (Nc), is an effective indicator for evaluating crop nitrogen levels [7]. Critical nitrogen concentration is defined as the minimum nitrogen concentration required to achieve maximum growth [8]. It is well known that nitrogen concentration in most crops decreases as biomass increases. This decrease in nitrogen can be described by the negative power of the dilution curve. Such a critical Nc dilution curve can be used to analyze the nitrogen nutrient status of crops at different growth stages, from which the NNI can be calculated [9]. With the deepening of research, further developments and applications at the organ scale have been carried out using the traditional Nc dilution curve. For example, in crops such as rice [10], winter wheat [11], and rape [12], the Nc dilution curve based on leaf dry matter (LDW) has been used as an effective indicator for the evaluation of nitrogen nutritional status. Using the relationship between dry matter and nitrogen concentrations in different parts of the crop, rather than using plant dry matter (PDM) alone, will be more conducive to improving nitrogen use efficiency and optimizing nitrogen management to achieve maximum crop yield goals [13]. As the main organ of metabolism and photosynthesis during crop growth, leaves play a decisive role in crop growth and yield [14]. The latest research on rice by Ata-Ul-Karim et al. [12] shows that leaves have the highest nitrogen use efficiency during the whole growth of rice, and the leaf-based dilution curve can better assess the nitrogen nutritional status of crops. At the same time, in terms of quantitative diagnosis of crop nitrogen nutrition, the NNI is more representative than a single indicator (e.g., leaf area, biomass, nitrogen content) because the NNI includes two crop group indicators (dry matter weight and nitrogen concentration) simultaneously [15,16]. However, using traditional methods to calculate the NNI is time-consuming, labor-intensive, and destructive, impeding the rapid calculation of the crop NNI [17]. The traditional method first requires destructive sampling in the field and then sending the sample to a laboratory for measurement of the nitrogen content of the crop through chemical analysis methods (such as the Kjeldahl method), before finally calculating the NNI through the formula. Although the results of this method are more accurate and reliable, the whole process is not only complex and time-consuming, but also consumes more resources and labor, which cannot effectively meet the requirements of rapid diagnosis of crop nitrogen nutritional status [2]. In contrast, the new hyperspectral method can provide a non-destructive, efficient, and rapid estimation of plant growth nutritional status [18], from which the plant NNI can be calculated.

As an alternative tool, spectroscopic equipment can assess many spectral wavelengths in the electromagnetic spectrum range, allowing the adequate selection of the effective wavelength and sensitive spectral index of different crop variables [19,20]. Studies have shown that spectroscopic techniques can be used to effectively estimate the crop NNI. For example, Mistele et al. [21] demonstrated that the correlation between the wheat NNI and canopy reflection intensity was stronger than that between the wheat NNI and nitrogen content or dry matter. Zhao et al. [22] used the newly developed two-band vegetation index to estimate the NNI of summer maize and demonstrated that the direct method is more effective than the indirect method. Thus, it is very important to use the correct method to extract useful information from the spectral reflectance to determine the nitrogen nutritional status of crops [23]. However, owing to the high similarity between adjacent bands in the raw hyperspectral data, there are considerable amounts of redundant information and irrelevant variables in the raw data, resulting in inaccurate prediction results and excessive calculation. Therefore, appropriate methods are generally adopted to screen the effective bands before further analysis and modeling. There are many methods used to select wavelengths for reflection spectra. The main methods include principal component

analysis, random leapfrog, the continuous projection algorithm, and competitive adaptive reweighted sampling (CARS) [24].

In this context, machine learning (ML) has developed rapidly as a branch of artificial intelligence (AI). ML can obtain useful information from a large amount of spectral data through self-learning to enable effective classification and self-prediction [25–27]. Shu et al. [24] achieved an accurate estimation of four traits (aboveground biomass, total leaf area, leaf chlorophyll content, and thousand kernel weight) in maize inbred lines using UAV hyperspectral images combined with two machine learning methods: partial least squares and random forest. Additionally, Li et al. [28] showed that the evaluation performance of a method based on machine learning, in particular the artificial neural network (ANN), was significantly better than that of traditional multiple linear regression. Fu et al. [29] analyzed existing studies and showed that support vector machine (SVM) regression, a kernel-based machine learning algorithm, has become an alternative to ANN in the evaluation of crop nitrogen nutrition status.

Maize, one of China's most important food crops, is also the main forage for animal husbandry [30]. Excessive nitrogen input in maize production reduces agricultural production efficiency and exacerbates ecological problems [31,32]. Therefore, a timely and accurate grasp of the nitrogen nutritional status during crop production is critical to the sustainable development of agriculture and achieving on-demand and precise nitrogen application [6]. To optimize the precise nitrogen management of field crops, increase crop yield, and improve the ecological environment, it is necessary to develop a fast and reliable method to determine the nitrogen nutritional status of maize crops. Until now, most crop NNI assessment methods have focused on the canopy level, and few studies have used in situ spectroscopy to estimate leaf nitrogen nutrition. Because crops are affected by environmental factors and canopy structure complexity, existing canopy-scale methods for crop nitrogen nutrition diagnosis suffer from low estimation accuracy and poor model migration, which indicates that further research is needed to improve the accuracy and universality of nitrogen nutrition monitoring. Using in situ leaf spectral methods to obtain leaf reflection data could effectively avoid the interference of canopy structure and environmental factors [2], improve the accuracy and effectiveness of leaf spectral reflection data, and provide an effective means for crop nitrogen nutrition assessment and field nitrogen management.

In this context, the present study presents a method for estimating the NNI of maize by combining different cultivars and nitrogen fertilizer treatments to accurately estimate maize nitrogen nutrition status. The main objectives were as follows: (1) to construct the best model combination of maize growth stages from the 12th fully expanded leaf stage to the milk ripening stage (V12–R3) by comprehensively analyzing the in situ reflectance spectra of leaves from visible light to near-infrared; (2) to compare the regression models of three regression methods—partial least squares (PLS), artificial neural network (ANN), and support vector machine (SVM)—on the raw hyperspectral bands and the effective bands; (3) to evaluate the performance of the combination of different methods and establish the most suitable maize NNI estimation model. The results of this study provide a technical basis for the application of hyperspectral technology in nitrogen nutrition monitoring and precise nitrogen application in maize production.

## 2. Materials and Methods

### 2.1. Experimental Site and Experimental Design

The experimental site was located at the Beijing Academy of Agriculture and Forestry Sciences facility in Tongzhou (116°41′2″ E, 39°41′50″ N) (Figure 1b), Beijing, China. This area has a typical warm temperate semi-humid continental monsoon climate. The soil type of the experimental field is sandy. Before maize planting, soil samples of 0–20 cm were collected to measure soil organic matter, total nitrogen, Olsen phosphorus, and available potassium. The soil properties are summarized in Table 1.

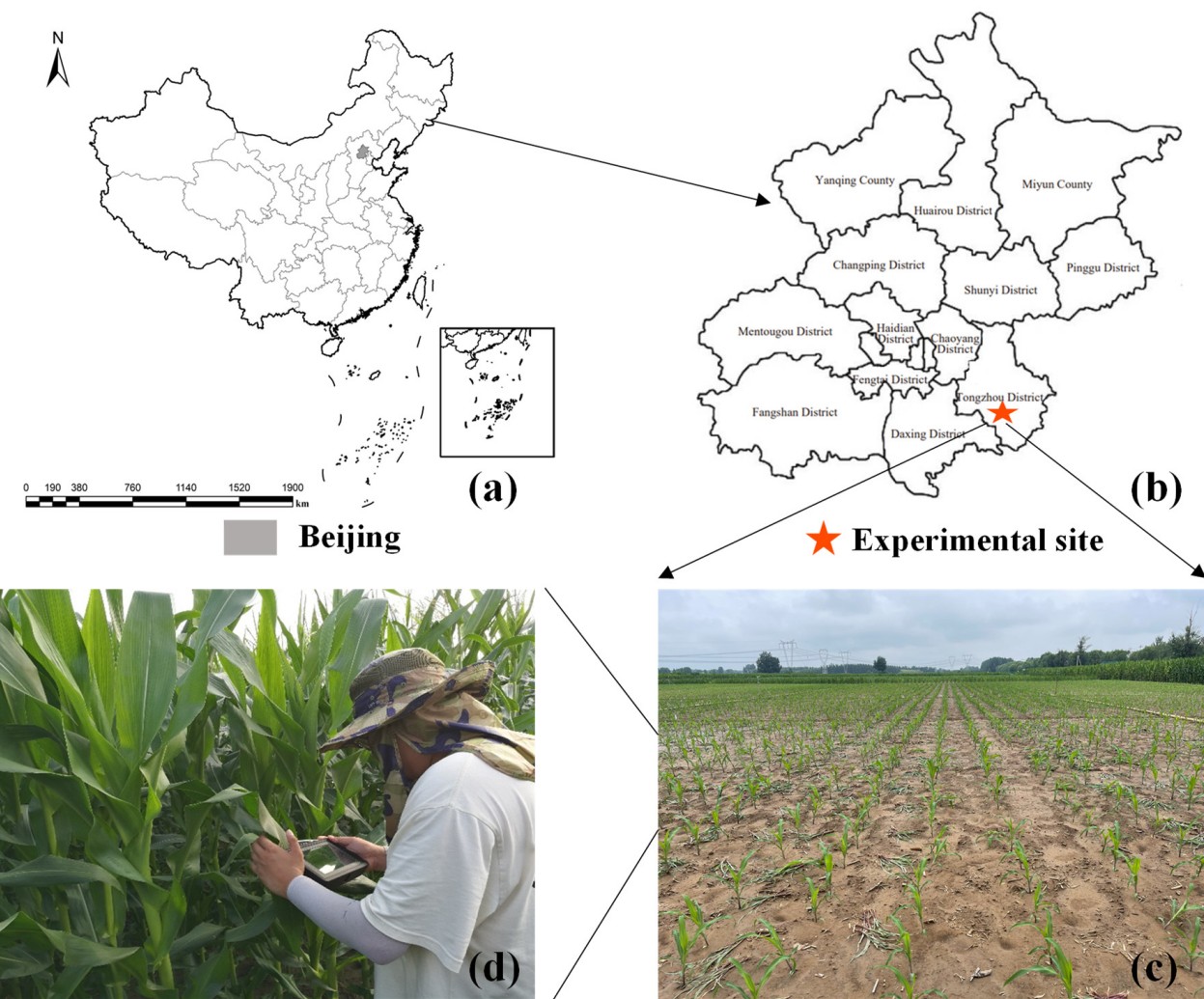

**Figure 1.** Location of the study site (**a,b**), field plot conditions (**c**), and leaf spectral reflectance measurements (**d**).

**Table 1.** Experimental design and main soil chemical properties.

| Cultivar | N Application (Kg N ha$^{-1}$) | Sowing/Harvesting Date | Sampling Stage | Soil Characteristics |
|---|---|---|---|---|
| Jingke999 (JK999) | | | | |
| Xianyu335 (XY335) | 0(N0) | | V6 | |
| MC121 | 75(N1) | | V12 | Type: brown sandy |
| Jingnongke728 (JNK728) | 150(N2) | 1 June | R1 | Organic matter: 17.03 g kg$^{-1}$ |
| Liangyu99 (LY99) | 225(N3) | 30 September | R3 | Total N: 1.08 g kg$^{-1}$ |
| MC812 | 300(N4) | | R5 | Olsen-P: 0.067 g kg$^{-1}$ |
| Jingnongke828 (JNK828) | 375(N5) | | R6 | Available-K: 0.241 g kg$^{-1}$ |
| Zhengdan958 (ZD958) | | | | |

This study adopted a split-plot experimental design: the nitrogen treatment was the main plot factor, and the cultivars were randomly distributed in each nitrogen treatment and repeated three times. Five field experiments were conducted to obtain maize leaf samples at the following growth stages: the 6th fully expanded leaf stage (V6), the 12th fully expanded leaf stage (V12), the silking stage (R1), the milk ripening stage (R3), the dent stage (R5), and physiological maturity (R6). This experiment included eight maize cultivars and six nitrogen application treatments (Table 1). Before sowing, 50% nitrogen fertilizer was applied as the base fertilizer, and a second application was carried out at the

jointing stage. Moreover, before sowing, all experimental treatments were given a one-time application of 90 kg ha$^{-1}$ superphosphate (P$_2$O$_5$) and 120 kg ha$^{-1}$ potassium oxide (K$_2$O). A total of 48 plots (30 m$^2$ each, 3 m × 10 m) were planted using the equidistant method with 60 cm row spacing, and the density was 60,000 plants ha$^{-1}$. In addition, the artificial sowing method was adopted, and the seedlings were thinned. The field was thereafter managed according to local practices.

### 2.2. Data Collection

#### 2.2.1. Leaf Hyperspectral Data Measurement

Spectral data of fully expanded leaves at the top of the maize shoot and leaves at the ear position were obtained from growth stages V12, R1, and R3, respectively (Figure 1d). Leaf reflectance data were measured using a leaf spectrometer CI-700s (CID Bio-Science, Camas, WA, USA), with a spectral range of 360–1100 nm, sampling speed of 3.8 ms to 10 s, light deviations of <0.05% at 600 nm and 0.10% at 435 nm, resolution of 0.55–0.7 nm, and full width at half maximum (FWHM) of 2.4 nm. The spectral reflectance results of leaves have low signal-to-noise ratios in the spectral ranges of 400 nm to 460 nm and 1000 nm to 1100 nm, so these bands were excluded from this experiment. Additionally, 460 nm to 1005 nm was used as the raw hyperspectral band for subsequent analyses. To ensure the use of the same parameters in different growth stages of maize and reduce the influence of environmental factors and the instrument itself, ten maize plants were selected from each plot and five parts of each leaf were selected for measurement and average value calculation. Sampling was conducted from 10:00 a.m. to 12:00 p.m. Beijing time, with calibrations every 5 min (100% reflectance was calibrated using the integrated BaSO$_4$ white standard, 0% reflectance was calibrated using the black standard), and ten spectra were selected per measurement. The average of ten spectral results was selected for each measurement for further study.

#### 2.2.2. Crop Biophysical and Biochemical Variable Measurement

Five maize plants were obtained at six growth stages to determine the leaf biomass (LDM) and leaf nitrogen content (LNC) in each plot, among which five maize plants were obtained from the V12–R3 stages, totaling three stages of leaf spectrum collection. The sampling period is shown in Table 1, and the aboveground maize was divided into different organs (stem, leaf, and ear portions) at each sampling period. The leaf samples were dried at 105 °C for 30 min to inactivate the enzymes and dried at 80 °C to a constant dry weight, and the LDM (t ha$^{-1}$) was then measured. Dried samples were ground, passed through a 1 mm sieve, and stored in paper bags for chemical analysis. All dried and ground samples were digested with H$_2$SO$_4$-H$_2$O$_2$ according to the method proposed by Thomas et al. [33], and then maize leaf nitrogen concentrations were determined using a flow injection autoanalyzer (AA3, Bran and Luebbe, Norderstedt, Germany).

### 2.3. NNI Calculation

Maize leaf Nc was determined using Equation (1) [8]. In Equations (1) and (2), W is the LDW, Nc is the leaf critical nitrogen concentration, and Na is the actual leaf LNC. Moreover, a represents the leaf nitrogen concentration at an LDM of 1 t ha$^{-1}$ and b is the curve's dilution factor. Therefore, the NNI can directly reflect the nitrogen nutrient status of plants. If the NNI > 1, the plant has excess nitrogen nutrition; if the NNI = 1, the plant's nutritional status is optimal; if the NNI < 1, the plant is deficient in nitrogen [34].

$$Nc = aW^{-b} \tag{1}$$

$$NNI = \frac{Na}{Nc} \tag{2}$$

### 2.4. Effective Band Selection

In this study, the CARS algorithm was used to screen the spectral data of maize leaves. CARS is a recently proposed variable selection method [35]. It combines Monte Carlo sampling (MC) and the regression coefficient of the PLS model. In addition, it uses a variable selection method according to the "survival of the fittest" principle based on Darwin's theory of evolution by nature and selects variables with a larger absolute coefficient in the multiple linear regression model. Finally, a cross-validation method is used to retain the subset with the smallest cross-validation root mean square error, which is the combination of the effective wavelengths. Such an application of hyperspectral data has proven to be very effective, as it can avoid overfitting when selecting variables for modeling and thereby improve a model's predictive ability [36,37].

After obtaining maize leaf hyperspectral data and ground data with a leaf hyperspectrometer, the effective band of the raw spectral bands was filtered using CARS. Then, the reflectivity of the effective band was used as an input variable. Next, PLS, ANN, and SVM were used to estimate the maize leaf NNI and compare the performance of the three models. Finally, three machine learning regression procedures were run using Matlab2020b (Math-Works, Natick, MA, USA).

### 2.5. Model

#### 2.5.1. Partial Least Squares Regression

Partial least squares regression (PLS) is a commonly used spectral data modeling method. It is suitable for analyzing multicollinear spectral datasets and high-dimensional data and can effectively solve the problem of the number of independent variables exceeding the number of samples [38]. At the same time, PLS regression has been demonstrated to be a general multivariate statistical regression method for modeling crop biochemical components using spectral data [2]. It successfully combines multiple linear regression, principal component, and correlation analyses. It can better solve the multicollinearity problem between variables [39].

#### 2.5.2. Artificial Neural Network Algorithm

Artificial neural networks (ANN) are an important advance in artificial intelligence. After years of research and development, methods based on ANN have played an important role in remote sensing, image recognition, and information retrieval. ANN regression is a nonparametric linear model that uses neural network layered propagation to simulate the reception and information processing of the human brain. Neural network regression is a gradient-based learning method that includes input, hidden, and output layers and network initialization [40]. The final weights are obtained by continuously updating the error values and weights. Previous studies have shown that ANNs are suitable for regression modeling data types with large sample sizes [41]. However, ANNs are susceptible to overlearning owing to the network structure and sample complexity, which reduces generalizability. Thus, neurons are an important parameter in neural network regression models. The more neurons, the higher the accuracy of the model and the weaker its generalizability.

#### 2.5.3. Support Vector Machine Algorithm

Support vector machine (SVM) methods are a machine learning technique [42], and support vector regression (SVR) is an important SVM application. Its principle is to find an optimal hyperplane that minimizes the total deviation of all sample points from the hyperplane and then fits all the data through the optimal hyperplane. In SVR, there are four kernels: linear, polynomial, Gaussian, and sigmoid [43]. In this study, the linear kernel was used, and its model performance is affected by the kernel parameter (gamma) and adjustment parameter (C).

*2.6. Data Analysis*

2.6.1. Training and Validation Datasets

A total of 144 samples were collected during the 2021 growing season, including three measurement periods (V12, R1, and R3, with 48 samples each) and six nitrogen fertilizer treatments (N0, N1, N2, N3, N4, N5). The entire sample dataset (144) was randomly divided into a training set (96) and a validation set (48) according to a 2:1 ratio.

2.6.2. Statistical Analysis

In this experiment, the coefficient of determination ($R^2$) and the root mean square error (RMSE) were used to quantify the amount of variance explained between the established relationships and the model's accuracy. In general, the stability of the model is assessed by comparing the difference in $R^2$ and RMSE. $R^2$ represents the fit between the predicted value and the measured value. The higher the $R^2$ value, the higher the accuracy of the model prediction. RMSE represents the degree of deviation between the predicted value and the measured value. The smaller the RMSE value, the higher the predictive accuracy of the model. The calculation formulas are as follows:

$$R^2 = 1 - \frac{\sum_{i=1}^{n}(y_i - P_i)^2}{\sum_{i=1}^{n}(y_i - \overline{y})^2} \tag{3}$$

$$\text{RMSE} = \sqrt{\frac{\sum_{i=1}^{n}(y_i - P_i)^2}{n}} \tag{4}$$

Here, $P_i$ represents the predicted value of the regression model, $\overline{y}$ represents the average value of the measured value, $y_i$ represents the measured value, and $n$ represents the sample size.

**3. Results**

*3.1. Construction of Critical Nitrogen Concentration Dilution Curve Based on Maize LDM*

This study selected eight summer maize cultivars to determine the Nc dilution curve based on maize LDM. The LDM and LNC value ranges were 0.16–2.78 t ha$^{-1}$ and 0.77–4%, respectively (Figure 2). The data were used to calculate the leaf Nc value at each growth stage. According to the three-step method proposed by Justes et al. [44], the LDM Nc dilution curves were constructed for every cultivar. The values of the LDM Nc dilution curves *a* (2.24–3.13%) and *b* (0.2–0.41%) and the $R^2$ (0.699–0.861) for the eight studied cultivars are shown in Table 2. The LDM dilution curves of the eight maize cultivars fit well and are representative. At the same time, the difference in *a* and *b* values between different curves is small. To further study the comprehensive evaluation model suitable for nitrogen nutritional status among multiple agricultural populations, the data from all cultivars were combined and a unified Nc dilution curve was fitted according to the following equation (Figure 3):

$$\text{Nc} = 2.748\text{LDM}^{-0.296} \tag{5}$$

**Table 2.** Dilution curve of the leaf dry matter (LDM) critical nitrogen concentration in maize between different cultivars.

| Cultivar | *a* (%) | *b* | $R^2$ |
|----------|---------|-----|-------|
| JK999    | 2.83    | 0.34 | 0.705 |
| XY335    | 2.62    | 0.41 | 0.802 |
| MC121    | 2.94    | 0.29 | 0.702 |
| JNK728   | 2.78    | 0.36 | 0.820 |
| LY99     | 2.97    | 0.20 | 0.804 |
| MC812    | 3.13    | 0.21 | 0.699 |
| JK828    | 2.68    | 0.38 | 0.857 |
| ZD958    | 2.24    | 0.35 | 0.861 |

Note: *a* is the Nc concentration for leaf dry matter (LDM) equal to 1t ha$^{-1}$ and *b* is the decline in Nc concentration with crop growth. $R^2$ is the fitting accuracy of the curve.

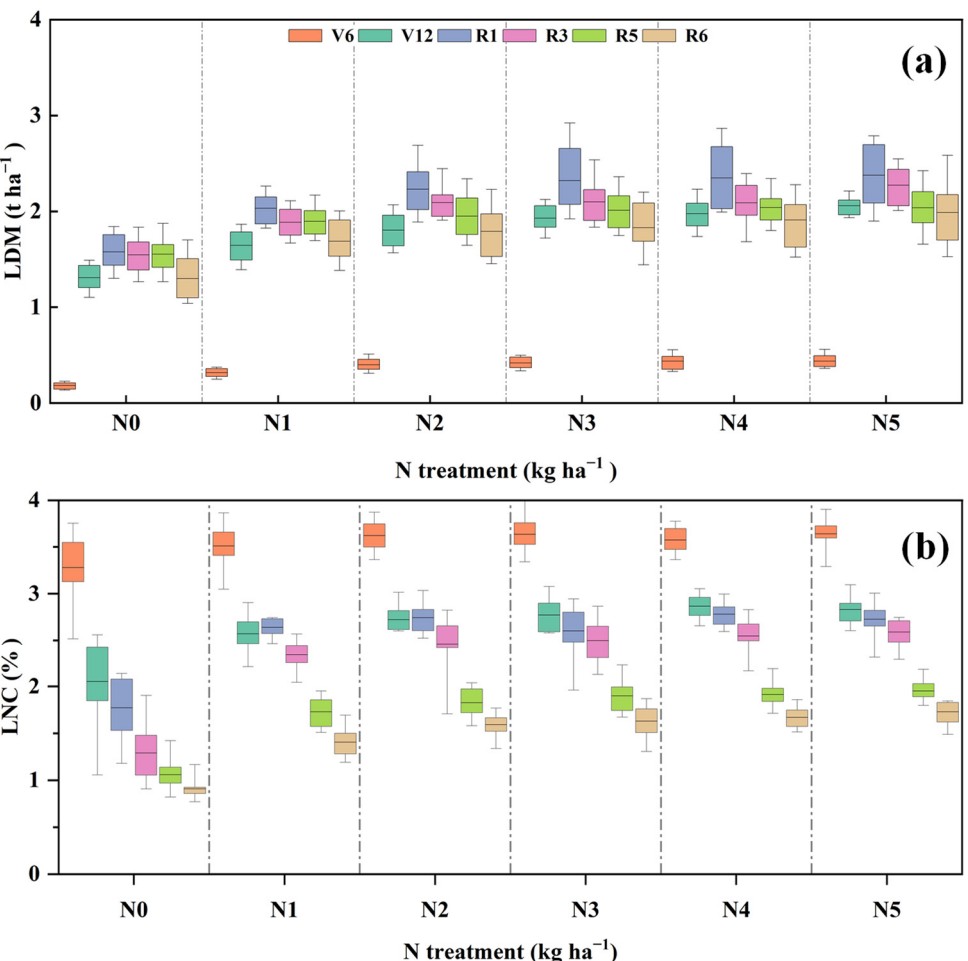

**Figure 2.** Changes in maize leaf dry matter (LDM) (**a**) and leaf nitrogen content (LNC) (**b**) at different growth stages under different nitrogen application rates. Note: N0–N5 represents different nitrogen application rates (0, 75, 150, 225, 300, and 375 kg N ha$^{-1}$); V6-R6 represents different growth stages (the 6th fully expanded leaf stage, the 12th fully expanded leaf stage, the silking stage, the milk ripening stage, the dent stage, and physiological maturity).

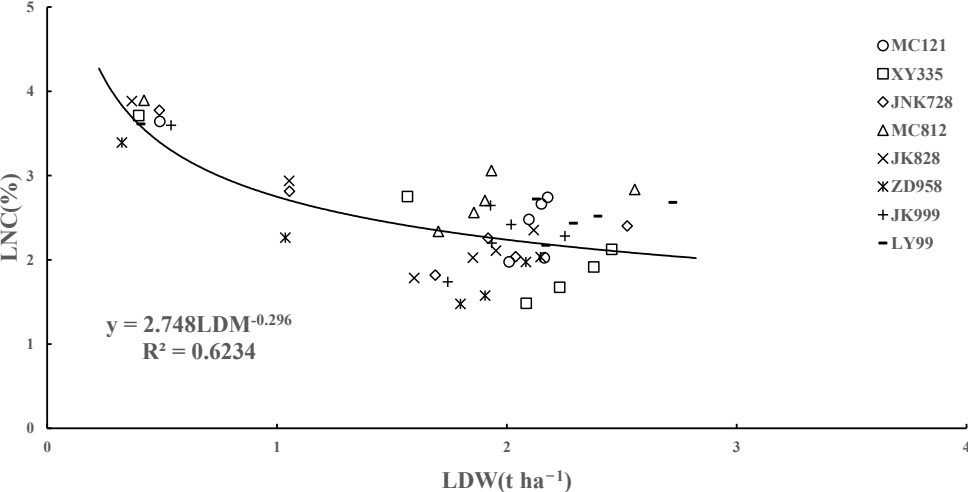

**Figure 3.** Summarized data from eight cultivars used to define the Nc dilution curve. Different symbols denote Nc data points for different cultivars. The solid line represents the Nc dilution curve (Nc = 2.75 LDW$^{-0.3}$, $R^2$ = 0.62), expressing the relationship between leaf nitrogen content (LNC) and leaf dry matter (LDM).

### 3.2. Leaf NNI

#### 3.2.1. Statistical Results of Maize Leaf NNI Dataset

The NNI determination results are summarized in Table 3. The maximum and minimum values of the NNI in the training set and validation set were similar and had similar ranges, indicating that the division of the dataset was reasonable. The training set NNI ranged from 0.369 to 1.395, with a mean of 1.110 and a standard deviation of 0.228. The validation set range, mean, and standard deviation were 0.479–1.400, 1.123, and 0.226, respectively.

**Table 3.** Statistical overview of the maize leaf NNI dataset.

| Sample Datasets | Number of Samples | Mean | Max [a] | Min [b] | SD [c] |
|---|---|---|---|---|---|
| Entire dataset | 144 | 1.110 | 1.400 | 0.369 | 0.228 |
| Training dataset | 96 | 1.103 | 1.395 | 0.369 | 0.228 |
| Validation dataset | 48 | 1.123 | 1.400 | 0.479 | 0.226 |

[a] Max: maximum; [b] Min: minimum; [c] SD: standard deviation.

#### 3.2.2. Dynamic Changes of Maize NNI under Different Nitrogen Application Conditions

During V12–R3, the NNI value of maize leaves increased with nitrogen application (N0–N5) (Figure 4). Moreover, the leaf NNI value showed a dynamic change: a parabolic trend across the three key growth periods. The maximum for the six nitrogen fertilizer treatments was reached at the R1 stage. The leaf NNI values of the V12, R1, and R3 stages were 0.498–1.380, 0.616–1.400, and 0.369–1.316, respectively. The average leaf NNI of all cultivars was lower than 1 in the three growth stages without nitrogen application (N0), indicating that maize growth without nitrogen application was affected by stress (Figure 4). The NNI gradually increased with nitrogen concentration, reaching its maximum under the N5 treatment. At the same time, owing to differences among cultivars, the NNI values of individual cultivars under N1 and N2 treatments were less than 1, indicating that different cultivars have different levels of nitrogen use efficiency and low nitrogen tolerance. However, the overall trend was the same.

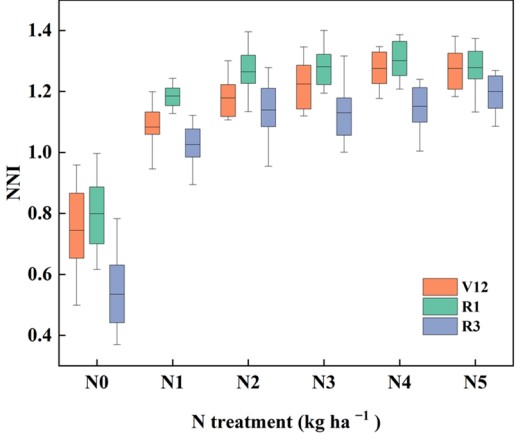

**Figure 4.** Dynamic changes of maize nitrogen nutrient index (NNI) under different nitrogen application treatments among the three growth stages.

### 3.3. Dynamic Changes in Maize Leaf Spectrum under Different Nitrogen Application Conditions

The spectral reflectance of leaves under different nitrogen fertilization treatments significantly differed. However, there were similar trends in different growth stages (V12, R1, and R3) (Figure 5). In the visible light band, the spectral reflectance of the N0 treatment was higher than that of the nitrogen application, but there was no significant difference among the different nitrogen application levels (N1–N5). In the near-infrared band, the spectral reflectance increased with the increase in nitrogen application rate, and the spectral

reflectance difference between different nitrogen application rates was greater in the near-infrared band than in the visible light band. Compared with the visible light band, the spectral reflectance in the near-infrared band was more sensitive to the nitrogen application rate. These results aid in establishing the quantitative relationship between the leaf NNI and leaf spectral reflectance features.

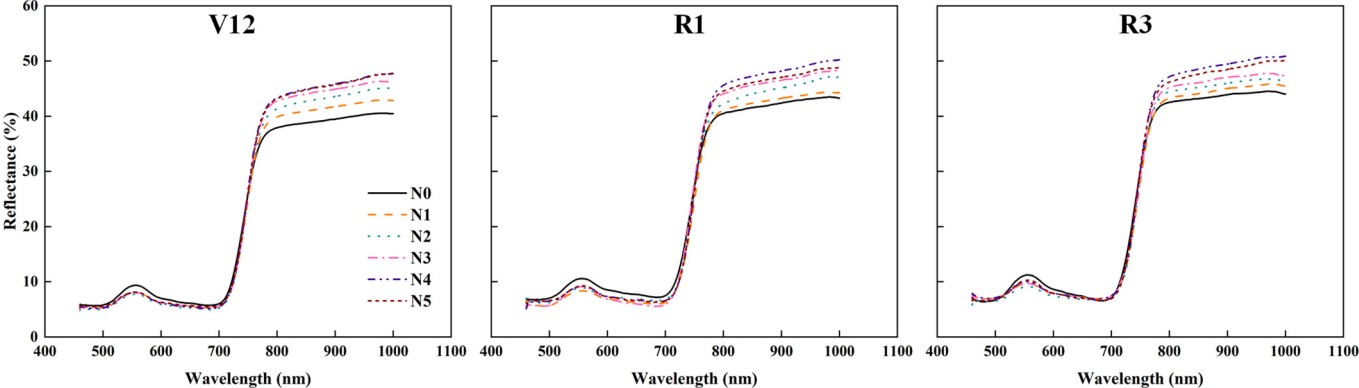

**Figure 5.** Changes in spectral reflectance of maize leaves across different nitrogen application rates (N1–N5) and different growth stages (V12–R3).

### 3.4. NNI Estimation

3.4.1. Maize NNI Estimation Model Based on Full-Band Reflectance

Table 4 summarizes the results of the NNI regression model of maize leaves constructed based on full-band reflectance. Three machine learning methods (PLS, ANN, and SVM) were used to construct NNI and full-band reflectance regression models of maize leaves. Then, the validation dataset was used to estimate the leaf NNI quantitatively. For the training dataset, the $R^2_{train}$ and RMSE$_{train}$ of the three regression models built by machine learning were quite different, and the ranges of $R^2_{train}$ and RMSE values were 0.649–0.903 and 0.071–0.135, respectively. Thus, the three regression methods have certain differences in the estimation of the maize leaf NNI, and thus, the more appropriate method should be selected. As shown in Figure 6, the ANN regression method produced a relatively high $R^2_{train}$ (0.903) and a low RMSE$_{train}$ (0.071). The SVM regression method achieved the second-best results ($R^2_{train}$ = 0.887, RMSE$_{train}$ = 0.077). Nevertheless, based on the results in the validation dataset, the SVM regression method (ALL-SVM) achieved a high $R^2_{val}$ (0.689) and a low RMSE$_{val}$ (0.126). In contrast, the ANN regression method (ALL-ANN) performed the worst ($R^2_{val}$ = 0.622, RMSE$_{val}$ = 0.138), indicating that the model constructed by this method has low stability in estimating the leaf NNI. Based on a comprehensive analysis, the ALL-SVM method has the highest estimation accuracy and was the most stable model among the three machine learning methods. Compared with the other two methods, it is an ideal method for NNI estimation.

**Table 4.** Leaf NNI estimation results based on different band combinations and different regression methods (PLS, ANN, and SVM).

| Bands | Numbers | Method | Training Set | | Validation Set | |
|---|---|---|---|---|---|---|
| | | | $R^2$ | RMSE | $R^2$ | RMSE |
| All bands | 933 | PLS | 0.649 | 0.135 | 0.627 | 0.138 |
| | | ANN | 0.903 | 0.071 | 0.622 | 0.138 |
| | | SVM | 0.887 | 0.077 | 0.689 | 0.126 |
| CARS | 67 | PLS | 0.946 | 0.050 | 0.925 | 0.068 |
| | | ANN | 0.857 | 0.082 | 0.814 | 0.108 |
| | | SVM | 0.947 | 0.050 | 0.895 | 0.081 |

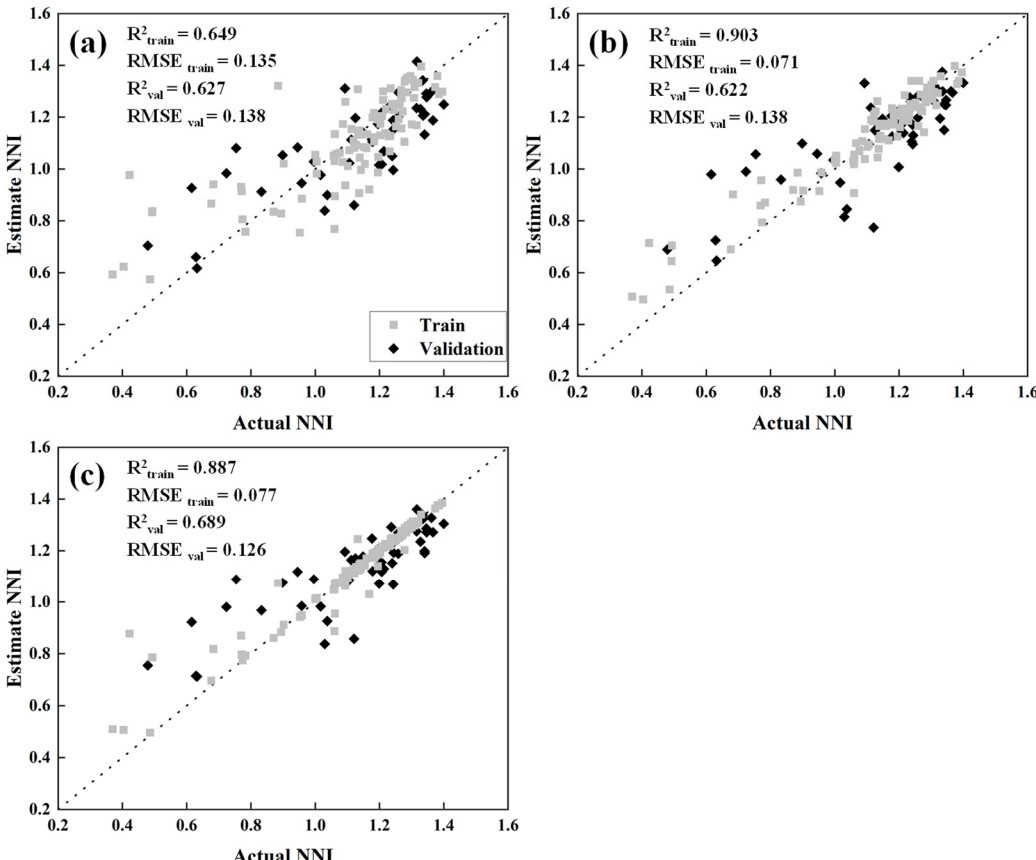

**Figure 6.** Comparison of the leaf nitrogen nutrient index (NNI) and full-band reflectance regression analysis results based on the following methods: partial least squares (PLS) (**a**), artificial neural network (ANN) (**b**), and support vector machine (SVM) (**c**).

3.4.2. Maize NNI Estimation Model Based on Effective Bands Reflectance

To further study the influence of different wavelength combinations on the estimation of the NNI of maize leaves, this study used the CARS algorithm to perform effective band screening on the full band of the raw spectrum. A total of 67 effective bands were selected from 933 raw spectral full bands (see Supplementary Material, Table S1). Table 4 lists the results of NNI models constructed with the PLS, ANN, and SVM methods. Figure 7 shows a scatter plot of the estimated NNI and actual NNI. NNI estimation accuracy and stability using effective bands are significantly improved compared to full-band modeling. It eliminates a large amount of invalid information and redundancy in the raw spectral data, greatly reduces the model calculation time, and significantly improves the model efficiency. The $R^2$ values of the training sets for the PLS, ANN, and SVM methods are 0.946, 0.857, and 0.947, respectively; the corresponding RMSE values were 0.050, 0.082, and 0.050, respectively. Among them, the $R^2$ values of the CARS-PLS and CARS-SVM methods were higher than 0.9. At the same time, the validation dataset based on the CARS-PLS method also obtained a relatively stable effect ($R^2_{val}$ = 0.925, RMSE$_{val}$ = 0.068), while the CARS-SVM method was slightly less effective ($R^2_{val}$ = 0.895, RMSE$_{val}$ = 0.081). Thus, the comprehensive analysis showed that CARS-PLS was the best method to estimate the NNI of maize leaves. Detailed parameter settings in the three models based on the full band and effective band are shown in Supplementary Material, Table S2.

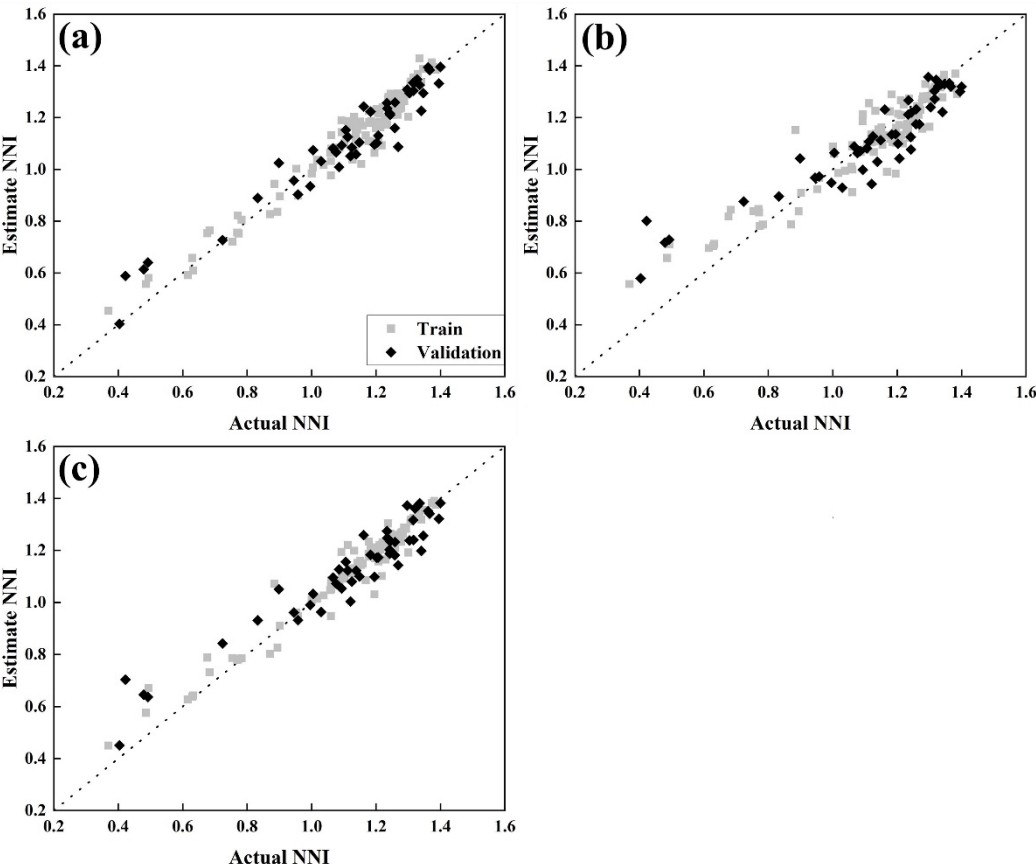

**Figure 7.** Comparison of the nitrogen nutrient index (NNI) and effective band (CARS) regression analysis results based on the following three methods: partial least squares (PLS) (**a**), artificial neural network (ANN) (**b**), and support vector machine (SVM) (**c**).

*3.5. Model Accuracy for Different Cultivars, Growth Stages, and Nitrogen Treatments*

The constructed NNI estimation model (CARS-PLS) was used to predict different cultivars, growth stages, and nitrogen treatments separately and further verify the efficacy of the model in NNI prediction of multiple cultivars and different treatments at different growth stages. The results are summarized in Figure 8. The datasets were treated according to nitrogen treatment (Figure 8a–f), cultivar (Figure 8g–n), and growth stage (Figure 8o–q) division. In this analysis, only the performance of the CARS-PLS method was evaluated, and the coefficient of determination ($R^2$) was used to evaluate the model. For N4 ($R^2 = 0.53$, Figure 8e) and N5 ($R^2 = 0.58$, Figure 8f), as the nitrogen application rate increased, the NNI was more concentrated in the region greater than 1, resulting in insufficient prediction ability of the model. However, under the low nitrogen treatment, the model was relatively stable (Figure 8a–d). At the same time, the model developed in this study has a good predictive ability for NNI prediction among cultivars (Figure 8g–n) and growth stages (Figure 8o–q). The range of $R^2$ among cultivars was 0.86–0.97, and the range of $R^2$ in the growth stages was 0.92–0.94. The NNI prediction model constructed using machine learning combined with spectral band selection method based on multi-growth stages data could realize NNI prediction among maize cultivars and multi-growth periods, but the performance was insufficiently stable for different nitrogen treatments.

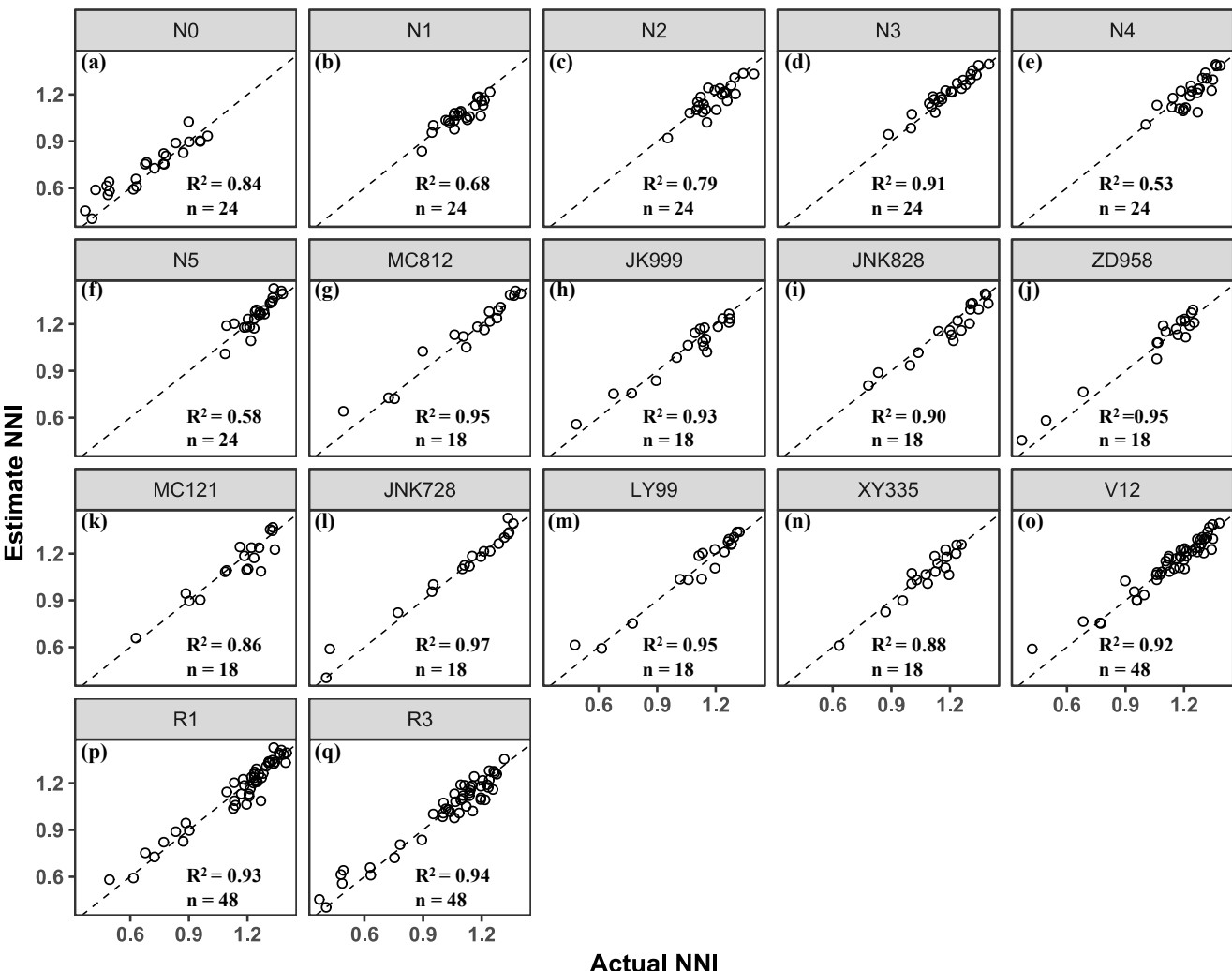

**Figure 8.** Comparison of the actual and estimated nitrogen nutrient index (NNI) for different nitrogen treatments (**a**–**f**), cultivars (**g**–**n**), and growth stages (**o**–**q**). Note: *n* represents the number of samples under each of the different conditions.

## 4. Discussion

### 4.1. Comparison with Other Nc Dilution Curves

The Nc dilution curve is a method for diagnosing the nitrogen nutritional status of crops based on the fast-growing relationship between crop nitrogen uptake dynamics and dry matter accumulation [16]. The Nc dilution curve constructed in this study based on the dry matter of the leaves of summer maize of various cultivars was as follows: Nc = $2.75(LDM)^{-0.3}$. As the maize crop grew, the leaf Nc value began to decrease slowly. Similar downward trends have been previously published for leaf nitrogen dilution curves of crops such as rice, canola, winter wheat, and maize [10,11,45,46]. The value of parameter a (2.75) estimated in the analysis was lower than that estimated by Zhao et al. based on the dry leaf matter of summer maize (Zhengdan958 and Denghai605) (a = 3.45) [15]. This difference may be associated with variations in experimental materials or different environmental and climatic conditions at the experiment site. The parameter of b represents the continuous decrease of LNC with the growth of the crop, and its value determines the rate of decrease of the curve. The b value of this study (0.3) was lower than the values estimated by previous studies for whole maize plants (0.37, 0.39, and 0.48) [8,47,48]. This difference shows that the rate of nitrogen dilution in maize leaves is slower than that in whole plants because leaves are the main site for photosynthesis, respiration, and

transpiration. Hence, a large amount of nitrogen inside maize is transferred from the stem to the leaves, resulting in a slow decline in LNC [15,46,49].

In addition, many researchers have developed various nitrogen concentration dilution curves for maize based on different ecological points and crops [47,48,50,51]. The purpose is to find a more accurate and reasonable evaluation system of maize nitrogen nutrition that aligns with the actual production conditions. In this study, the nitrogen dilution curves constructed using eight different maize cultivars and six nitrogen fertilization treatments can better reflect the current situation of nitrogen nutrition during maize growth. Therefore, the nitrogen dilution curve based on LDM provides a reference value for the in-depth understanding of the nitrogen nutritional status of maize and lays a foundation for future non-destructive monitoring of the maize NNI.

### 4.2. Response of NNI to Leaf Spectra

As a diagnostic indicator of crop nitrogen nutrient status, the NNI includes two group indicators: dry matter weight and nitrogen concentration. Thus, it has been considered a current and reliable method for diagnosing plant nitrogen status, which provides an effective way to ensure food production and quality [22,44]. However, the experimental data of this study covered different nitrogen treatments, growth stages, and cultivars, which gave the experimental data wide variation. The training set NNI range was 0.369–1.395, and the validation set NNI range was 0.479–1.400. Thus, there are differences in the datasets (Table 3). Notwithstanding, this result is similar to those of Kokaly and Marten et al. [52,53], demonstrating that the wide variation of the present dataset helps to build stable models.

In addition, the reflectance of maize leaves showed a regular change under different nitrogen application conditions (Figure 5). The spectral reflectance of all nitrogen treatments had similar response patterns. With the increase in nitrogen application rate, the reflectivity in the near-infrared region (750–1000 nm) gradually increased. In the visible light region (460–710 nm), the reflectance of leaves under no nitrogen application was higher than that under nitrogen application (Figure 5), and there was no significant difference between nitrogen application treatments. This result is consistent with the findings of Zhao et al. [22]. Barłóg et al. showed that an adequate supply of nitrogen fertilizers promoted crop growth and increased leaf greenness [54]. The opposite results were shown in the near-infrared wavelength range, where there was a clear difference between different nitrogen application rates owing to internal scattering in the leaves, in agreement with the findings of Yao and Walburg et al. [55,56]. These results provide a theoretical basis and technical route for the rapid and accurate construction of the quantitative relationship between maize NNI and leaf reflectance spectra.

### 4.3. Optimal Model for Maize NNI Estimation

Choosing an appropriate method to estimate the NNI is critical for quantitative monitoring and inversion of crop nitrogen nutrient levels [24,57]. Under field conditions, the changes in leaf structure and physiological and biochemical indicators differed among growth stages, cultivars, and nitrogen treatments. These differences likely caused differences in spectral characteristics at the level of crop leaves. The present results show that the field in situ leaf spectroscopic method developed in this study can be applied to maize NNI estimation, obtaining relatively ideal results. The best monitoring model for NNI estimation was constructed using the CARS band selection method (Figure 7). Li et al. showed that CARS could eliminate invalid information variables or select effective bands, thereby establishing a model with stable performance and high accuracy [35]. Because the NNI is determined by both nitrogen content and biomass, the spectral features related to the NNI are closely related to these two traits [22]. In this study, a total of 67 spectral effective bands were identified by the CARS algorithm to estimate the maize NNI, and 13 wavelengths were identified in the range of visible light (400–710 nm), which is closely related to the absorption wavelengths of pigments in crops. There was a strong correlation between the nitrogen nutritional status of crops and chlorophyll concentration.

Therefore, the spectral characteristics of crop chlorophyll are directly related to nitrogen nutrition status [58,59]. The model identified 43 wavelengths in the near-infrared range. Zhao et al. have demonstrated that there is a sensitive spectral band related to the NNI in the near-infrared region, and there is a strong linear relationship between the NNI and the spectral index established by using the wavelength in the sensitive region [22]. At the same time, a total of 11 wavelengths were identified in the red edge region by this model. Owing to the strong absorption of chlorophyll at 680 nm and multiple canopy scattering at 780 nm, the reflectance at the red edge position abruptly increased. Read et al. found that the correlation between cotton leaf quantity and reflectance was highest in the red edge (700–710 nm) region [60]. Zhao et al. [61], by studying the relationship between nitrogen concentration and reflectance in sorghum leaves, showed that the first derivative of red reflectance centered at 730 or 741 nm obtained the best fitting accuracy, $R^2 = 0.73$. In summary, the CARS algorithm was used to screen out many invalid spectral variables and redundant information in the raw band data, and the effective information related to crop nitrogen nutrition was retained, which not only reduced the impact of invalid spectral information on the results but also effectively improved the processing efficiency of the model. Better prediction results are obtained by using meaningful partial bands rather than continuous bands or combinations of bands. Compared with full-band modeling (ALL-PLS, ALL-ANN, and ALL-SVM), modeling with effective bands after CARS screening yielded higher $R^2$ values (0.925, 0.814, and 0.895) and lower RMSE values (0.068, 0.108, and 0.081).

To further verify the stability of the model under different conditions, the experimental dataset was divided according to variety, growth period, and nitrogen treatment, and then the stability of the model was tested. The CARS-PLS model had a relatively ideal predictive ability among different varieties (Figure 8g–n), and the model accuracy $R^2$ value ranged from 0.86 to 0.97, indicating that the effect of the CARS-PLS model was not easily affected by variety differences. The validation results of the three independent growth period datasets also obtained ideal results, with $R^2$ values above 0.9 (Figure 8o–q), which basically met the needs of nitrogen nutrient level evaluation of field crops. Further tests are needed to verify whether this model is applicable to the early growth stage (jointing or seedling stage) of maize. In addition, for different nitrogen application treatments (Figure 8a–f), the accuracy of the model under N4 and N5 nitrogen application treatments was lower, with $R^2$ values of 0.53 and 0.58, respectively. This may be owing to the difference in nitrogen use efficiency among different cultivars, and the optimal nitrogen application rate of different cultivars differed.

Thus, the field in situ leaf spectroscopy method developed in this study can be applied to evaluate the nitrogen nutrient status of maize and has achieved good results among different varieties, during critical growth stages, and under different nitrogen fertilizer application levels. In actual production, determining the nitrogen nutritional status of crops accurately and quickly has always been an urgent problem for breeders and production managers. The in situ leaf spectroscopy developed in this paper provides a rapid, economical, and non-destructive method for the analysis of the nitrogen nutrient status of maize, offering a new technical means for the selection of nitrogen-efficient maize varieties and for informing nitrogen management in the field.

## 5. Conclusions

In this study on summer maize, the Nc dilution curve based on the dry matter of maize leaves was constructed according to the nitrogen dilution theory of crops. Thus, the maize NNI can be accurately estimated using in situ leaf spectral reflectance in the field. A total of 67 effective bands were identified by the CARS method. Combining effective bands and machine learning was found to improve the model's accuracy and operating efficiency. Among the three methods studied, the CARS-PLS method yielded higher $R^2$ values ($R^2_{val} = 0.925$) and lower RMSE values ($RMSE_{val} = 0.068$) between measured and estimated NNI compared to CARS-ANN and CARS-SVM values, indicating that CARS-PLS is the best method for NNI estimation. In addition, the CARS-PLS model also showed a

strong NNI estimation among cultivars ($R^2$, 0.86–0.97) and growth stages ($R^2$, 0.92–0.94). Although the model in this study was constructed using data collected from different cultivars, different growth stages, and different nitrogen application rates, data from only one growing season were obtained. Future studies should validate the model by applying it to a wider range of datasets, including different ecological environments, different growing seasons, and different crops, to further validate the performance and robustness of the method.

**Supplementary Materials:** The following are available online at https://www.mdpi.com/article/10.3390/agriculture12111839/s1, Table S1: Result of effective band screening based on CARS method. Table S2: Parameter setting of three regression models (PLS, ANN and SVM) based on different band combinations.

**Author Contributions:** Conceptualization, C.Z., X.G. and B.C.; Methodology, B.C., S.G., G.H., X.G. and C.Z.; Validation, S.Y. and G.H.; Formal analyses, S.Y., X.L. and S.G.; Investigation, B.C. and X.L.; Resources, C.Z. and X.G.; Data curation, B.C., G.H. and S.G.; Writing—original draft, B.C.; Writing—review and editing, X.L., X.G. and C.Z.; Funding acquisition, C.Z. and X.G.; Supervision, C.Z. and X.G. All authors have read and agreed to the published version of the manuscript.

**Funding:** This research was funded by (1) The National Natural Science Foundation of China (No.31871519); (2) Construction of Collaborative Innovation Center of Beijing Academy of Agricultural and Forestry Sciences (No. KJCX201917).

**Institutional Review Board Statement:** Not applicable.

**Data Availability Statement:** Not applicable.

**Conflicts of Interest:** The authors declare no conflict of interest.

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
