# Peer review of "The Application of Machine Learning Models Based on Leaf Spectral Reflectance for Estimating the Nitrogen Nutrient Index in Maize"

_agriculture, doi:10.3390/agriculture12111839_

Round 1

Reviewer 1 Report

The experimental work presented in the Manuscript, entitled „ Estimation of Maize Nitrogen Nutrient Index (Nni) Using In Situ Leaf Spectroscopy  " is interesting research with some promising results. The article reports , combining effective bands and PLS (CARS- PLS) was the best model for NNI estimation, which yielded the highest coefficient of determination (R2 val) of 0.925, and the lowest root mean square error (RMSE val) of 0.068. The CARS-PLS model was used to successfully predict the variation of NNI between breeds and different growth stages. The estimated R2 of eight cultivars by NNI was between 0.86 and 0.97; the estimated R2 of NNI atdifferent growth stages was between 0.92 and 0.94, there are several shortcomings and modifications that should be included in order to enhance the final manuscript for the readers.

1-      The title of the manuscript is traditional and should be modified. For example, the application of machine learning models based on spectral reflectance to estimate maize nitrogen nutrient index at leaf level.

2-      Abstract

·         Line 15 : please change We propose to the objectives this study to estimate NNI using …………… of this study.

·         Line 25 please remove this sentence (The leaf spectral reflectance was higher without nitrogen application in the visible band and lower in the near-infrared (NIR) band).

·         Line 26 please start the sentence by the results showed that …. Combining effective bands.

·         Which growth stage presented the best R2 and small RMSE asses NNI by machine learning models based on spectral reflectance?

·         Please arrange the efficiency of the three models in abstract to be clear for the readers?

3-      Introduction

·         The authors should present the previous work as possible about the (partial least squares – PLS, artificial neural network – ANN, and support vector machine – SVM) in this section.

·          Please highlight in introduction, what is the novelty (originality) of the work? And what is new in your work that makes a difference in the body of knowledge? What has been done that goes beyond the existing research.

·         Line 92:   The authors said that at canopy scale due to the influence of crop environmental factors and the complexity of the canopy structure, traditional monitoring methods cannot accurately reflect the spectral information of leaves. But also the estimate NNI at leaf level is time consuming and also can not cover large area of maize. Please present the advantage and disadvantage of your method.

4-       Materials and Methods

·         The materials and methods are well written

5-      Results as well as  discussion are well written

6-      Please, write the practical applications of your work in a separate section, before the conclusions and provide your good perspectives.

7-      Please write about the limitation of this work under the section of conclusion.

Reviewer 2 Report

The manuscript presents an insight on the most suitable method to be used in the estimation of maize nitrogen nutrient index (NNI) using in situ leaf spectroscopy. The findings presented in the study are interesting and well-articulated. However, few issues need to be addressed in order to bring the manuscript to an acceptable form. My specific comments are:

 1. The Nni should be capitalized in the title.

2. The authors need to explain the meaning of V12 and R3 in the abstract and in the introduction sections.

3. Write in third person, avoid personal pronouns, such as we, they, you, I, or our , their, yours.

4. Manuscript English should be improved to make optimal sense of the paper. It makes the paper unprofessional I am afraid. The authors need to find someone or a professional service to revise it.

5. Authors claimed in line 67 that “using traditional methods are time-consuming …..”, please explain what are the traditional methods used to estimate NNI, what are their advantages and disadvantages, and then explain the application advantages of the hyperspectral or spectroscopy method. Also, in lines 92-94, how you can justify your method (spectroscopy) being more accurate than the traditional monitoring methods mentioned here, and what are those traditional methods?  

6. Please add “to” to objective 3 to be consistent with your writing.

7. Fix the error in line 174.

8. Revised subtitle 2.5 and capitalize each word in subtitle 2.5.1 to be consistent with your writing.

9. Delete the last sentence in section 2.6.2 lines (243-244) as this is redundant information.

Reviewer 3 Report

In this paper, this manuscript was studied to predict the Maize Nitrogen Nutrient Index using 360-1100nm spectrum.There are several problems:

1. The range of spectral features are not introduced, thus the explanation of the spectral mechanism related to nitrogen element must be improved. Why using feature extraction methods are better than full-band models?

2. In P332, 3.4.2, there were not introducing the parameter setting and training process of models.What are the principal components of PLS, the neuronal layers of ANN and the kernel function of SVM?

3. Predicting NNI of  3.5, are analyzed during different stages. The CARS-PLS is used , but what are the characteristics of different stages? And this part is not appeared in the discussion and conclusion.

4. The contents of chapter 449 and 4.4 are strange, so it is suggested to delete or merge with 4.3

5. The Conclusion must be written and improved.

Reviewer 4 Report

In Manuscript Agriculture-1971159Estimation of maize nitrogen nutrient index (NNI) using in situ leaf spectroscopy” authors have proposed a non-destructive method for estimating maize's nitrogen nutrient index (NNI) using in situ leaf spectroscopy. Manuscript is well written and is of great significance for scientific community and common readers.

General Comments

·       Title: capitalize “NNI”

·       In Introduction section discuss Nitrogen Dilution Curve

·       Carefully check formulas for superscript and subscript

·       L-24: Revise the text

·       L41:smaller crop leaves and increased leaf area?

·       L-97: Breeding? Revise the text

·       L43-48 &  L98-104 can be merged

·       L120: Add China

·       Table 1 shift with 2.1 Experimental site and Experimental Design

·       Conclusion: Add brief recommendation/s

Round 2

Reviewer 1 Report

Many thanks, the manuscript was significantly improved.

Reviewer 3 Report

There are no more obvious problems and this manuscript could be accepted.